# Genome-Wide Association Studies for the Concentration of Albumin in Colostrum and Serum in Chinese Holstein

**DOI:** 10.3390/ani10122211

**Published:** 2020-11-26

**Authors:** Shan Lin, Zihui Wan, Junnan Zhang, Lingna Xu, Bo Han, Dongxiao Sun

**Affiliations:** 1Key Laboratory of Animal Genetics and Breeding of the Ministry of Agriculture, National Engineering Laboratory for Animal Breeding, Department of Animal Genetics and Breeding, College of Animal Science and Technology, China Agricultural University, Beijing 100193, China; shanlin0820@163.com (S.L.); cauzhangjn@163.com (J.Z.); Lingna_Xu@163.com (L.X.); bohan@cau.edu.cn (B.H.); 2Stae Key Laboratory of Agriobiotechnology, College of Biological Sciences, China Agricultural University, Beijing 100193, China; 15810283027@163.com

**Keywords:** albumin, Chinese Holstein, genome-wide association study, inflammation, SNP

## Abstract

**Simple Summary:**

The early death and illness of newborn calves result in enormous economic losses in the dairy industry. As the immune system has not been fully developed in neonates, the adequate intake of nutrients and immune substances in colostrum is essential for protecting neonates from infections in their early life. The term albumin refers to a group of multifunctional proteins that are important in anti-inflammatory and anti-oxidative reactions and can help calves against various infections. Albumin of bovine whey is derived from the serum; hence, the concentration of albumin in colostrum and serum could be important traits for the breeding of potential natural disease resistance in dairy cattle. Herein, genome-wide association studies (GWASs) were performed to identify the candidate genes associated with albumin concentrations in colostrum and serum to provide useful molecular information for the genetic improvement of disease resistance traits in dairy cattle.

**Abstract:**

Albumin can be of particular benefit in fighting infections for newborn calves due to its anti-inflammatory and anti-oxidative stress properties. To identify the candidate genes related to the concentration of albumin in colostrum and serum, we collected the colostrum and blood samples from 572 Chinese Holstein cows within 24 h after calving and measured the concentration of albumin in the colostrum and serum using the ELISA methods. The cows were genotyped with GeneSeek 150 K chips (containing 140,668 single nucleotide polymorphisms; SNPs). After quality control, we performed GWASs via GCTA software with 91,620 SNPs and 563 cows. Consequently, 9 and 7 genome-wide significant SNPs (false discovery rate (FDR) at 1%) were identified. Correspondingly, 42 and 206 functional genes that contained or were approximate to (±1 Mbp) the significant SNPs were acquired. Integrating the biological process of these genes and the reported QTLs for immune and inflammation traits in cattle, 3 and 12 genes were identified as candidates for the concentration of colostrum and serum albumin, respectively; these are *RUNX1*, *CBR1*, *OTULIN,*
*CDK6*, *SHARPIN*, *CYC1*, *EXOSC4*, *PARP10*, *NRBP2*, *GFUS*, *PYCR3*, *EEF1D*, *GSDMD*, *PYCR2* and *CXCL12*. Our findings provide important information for revealing the genetic mechanism behind albumin concentration and for molecular breeding of disease-resistance traits in dairy cattle.

## 1. Introduction

Improved animal health and resistance to pathogens is an increasingly important breeding objective in the dairy industry [1,2]. As the immune system of newborn calves is too weak to fight various infections, most diseases (for e.g., flu, diarrhea and omphalitis) and death events affecting calves occur in the first few days after birth [3,4,5]. Colostrum provides numerous nutrients, regulatory factors (cytokines, growth factors, enzymes and hormones), immune factors (immunoglobulins) and crucial proteins (albumin) to guarantee the health of the newborn calves [6,7]. Albumin is synthesized by hepatocytes and penetrates into the milk through the epithelial tight junction from the blood plasma [8,9,10,11]. The physiological and pathophysiological functions of albumin are relatively well known, such as maintaining the osmotic pressure and having an anti-oxidizing effect during inflammatory reactions [12,13,14]. In dairy cattle, the albumin concentration has been found to be much higher in the first milked colostrum (1.21 ± 0.44 mg/mL) when compared to reported milk levels of <0.2 mg/mL [15]. Previous studies have shown that the concentration of albumin in the milk increased during functional transitions from lactation to involution and during inflammation in cows [11,14,16,17,18,19,20,21,22], sheep [23] and goats [24]. During mastitis, the mammary gland is exposed to a high level of free radicals, and albumin might enhance the anti-oxidant defenses of the glands [13,25]. Therefore, the albumin concentration of colostrum and serum offers a potential possibility as one of the indices for resistance breeding to decrease the mortality of newborns in dairy cattle.

The concentration of albumin is a typical quantitative trait [26]. Previous studies in humans have reported heritability estimates of 0.10–0.24 and 0.30–0.39 for glycated and excreted albumin, respectively [27,28]. In dairy cattle, heritability is relatively lower (0.13 ± 0.09) [29,30]. In early-lactation dairy cows, serum albumin has moderate heritability (0.27 ± 0.06) [31].

A genome-wide association study (GWAS) is a practical approach for the high-resolution mapping of loci controlling quantitative traits and has been widely applied in domestic animals [32]. In dairy cattle, a large number of previous GWASs have been performed to detect the genetic markers, candidate genes and QTLs for milk yield, milk protein and fat [33,34,35,36], milk fatty acids [37], mastitis [34,38,39,40], reproduction [35], body conformation [35] and immunoglobulin concentration [36,41]. However, so far there is only a limited number of studies that have investigated the candidate genes for albumin concentration in dairy cattle or even in other species. In this study, we conducted GWASs to identify the significant single nucleotide polymorphisms (SNPs) and candidate genes for the concentration of colostrum and serum albumin and provide information for molecular breeding to improve the resistance or tolerance to pathogens of newborns in dairy cattle.

## 2. Materials and Methods

### 2.1. Animals and Phenotypes

Blood, colostrum and hair follicle samples were collected in the first milking within 24 h after calving from 572 Chinese Holstein cows (0–6 h: 481 cows; 6–12 h: 38 cows; 12–18 h: 39 cows; and 12–24 h: 14 cows). All of the cows were from 10 dairy farms of the Beijing Dairy Cattle Center and the Beijing Sunlon Livestock Development Company Limited (herd 1: 90 cows; herd 2: 5 cows; herd 3: 88 cows; herd 4: 92 cows; herd 5: 58 cows; herd 6: 47 cows; herd 7: 75 cows; herd 8: 56 cows; herd 9: 19 cows; and herd 10: 42 cows). Cows were in parity of 1 to 4 and were the offspring of 44 sires. Cows were 23–72 months old at the time of calving. The whole procedure for collecting the blood, colostrum and hair follicle samples was carried out in strict accordance with the protocol approved by the Animal Welfare Committee of China Agricultural University (permit number: DK996).

The concentration of albumin of every colostrum and serum sample was measured by commercial ELISA kits (Bovine Albumin ELISA Quantitation Set, E10-113, Bethyl Laboratories, Montgomery, TX, USA). Concentrations were log_10_-transformed to follow a normal distribution.

### 2.2. Genotypes and Quality Control

The extraction of genomic DNA from the hair follicles of the 572 cows was carried out using a QIAamp^®^ DNA Mini Kit (QIAGEN, Valencia, CA, USA). Then the extracted DNA was genotyped by a GeneSeek GGP_HDv3 chip (including 140,668 SNP markers: GeneSeek, Lincoln, Dearborn, MI, USA). 

Quality control was conducted on PLINK 1.90 software and the filtering processes were as follows: Firstly, samples with genotyping <95% of the SNPs were deleted; then, SNPs with call rates <90%, minor allele frequencies (MAF) <0.1 and Hardy–Weinberg equilibrium (HWE) *p*-values < 10^−6^ were discarded [42,43]. Thus, 563 individuals with 91,620 SNPs were kept for further analysis (Appendix A).

### 2.3. Statistical Analysis

The association analysis for each SNP was implemented independently for the albumin concentration of colostrum and serum with the following mixed linear model:y = μ + bX+fM + g + e
where **y** is the vector of the phenotype of the log-transformed corrected concentration of albumin for 563 cows; **μ** is the vector of the overall mean; **b** refers to fixed effects, including herd, parity, time from calving and season of calving; **X** is the incidence matrix of b to y; **f** is the vector of the additive effect of the candidate SNP to be tested for association; **M** is the vector of the genotypes for the SNP, coded as 0 = BB, 1 = AB and 2 = AA; g is the vector of the polygenic effect with g ~N (0, **G**σ_g_^2^), where **G** is the genomic relationship matrix between pairs of individuals from all of the SNPs’ 30 chromosomes, including the **X** chromosome [44]; σ_g_^2^ is the additive variance; and e is the vector of the residual effects with e~N (0, 0, **I**σ_e_^2^), where σ_e_^2^ is the residual error variance.

The GWASs were implemented with GCTA v1.90.2 software, which estimates the variance explained by all of the SNPs on a chromosome or on the whole genome for a complex trait [44]. As Bonferroni correction is deuced strict and may cause false negative results [45], we used the false discovery rate (FDR) method to set the *p*-value threshold [46,47]. The genome-wide threshold value was calculated according to an FDR of 0.01 with the following formula: P = FDR × n/l
where n is the number of the SNPs with *p* < 0.01 in the GWAS results for the concentration of colostrum or serum albumin, and l is total number of SNPs analyzed. Then, we calculated the genomic inflation factor λ value to evaluate the extent of the population stratification by GenABEL packages [48] in R 3.6.0 (http://www.R-project.org/) [49]. Quantile–quantile (QQ) and Manhattan plots were drawn using the qqman package [50].

### 2.4. Candidate Genes

To further identify the positional candidate genes for albumin, the genes that contained or were close to (±1 Mbp) the significant SNPs were selected based on the Ensembl *Bos taurus* UMD3.1 database (http://www.ensembl.org/index.html). The extracted genes were then submitted into DAVID Bioinformatics Resources (https://david.ncifcrf.gov) for the Gene Ontology (GO) terms and Kyoto Encyclopedia of Genes and Genomes (KEGG) pathway analysis to identify the biological processes that these genes are involved in. Simultaneously, we compared the physical position of these genes that contained or were adjacent to significant SNPs with the known QTLs for inflammation and immune-related traits in the Cattle QTL database (https://www.animalgenome.org/cgi-bin/QTLdb/BT/index).

## 3. Results

### 3.1. Phenotype and SNP Data Statistics

In this study, 572 animals with the phenotype of an albumin concentration in the colostrum and serum were included for analysis. The original phenotypic data were log_10_-transformed to accomplish normality (Figure 1). The mean and the corresponding standard deviations for the original and corrected phenotype of the albumin concentration are shown in Table 1. After the quality control, 91,620 SNPs and 563 cows were obtained for the subsequent association analysis. The distribution of the SNPs on the genome is presented in Figure 2.

### 3.2. Genome-Wide Association Study

According to the *p*-value of every SNP, the calculated inflation factor (λ) was 0.983 and 1.004 for the concentration of colostrum and serum albumin, respectively, and the QQ plots (Figure 3) showed that the population stratification had been well corrected [51]. Consequently, nine significant SNPs were identified for concentration of albumin in colostrum at the genome-wide level (FDR at 1%), located on *Bos taurus* autosome (BTA) 1: 149,246,858 bp (BovineHD0100043239); BTA 3: 118,061,120 bp (ARS-BFGL-NGS-75,987); BTA 17: 68,421,115 bp (ARS-BFGL-NGS-66,134); and BTA 20: 59,137,600—69,916,426 bp (BovineHD2000016546, BovineHD2000016866, BovineHD2000019816, BTB-00798071, ARS-BFGL-NGS-75636, and ARS-BFGL-NGS-114933) (Table 2).

For the albumin concentration in serum, seven genome-wide significant SNPs were detected (FDR at 1%) on BTA 4: 10,737,673 bp (Hapmap39425-BTA-70290); BTA 7: 93,597,405 bp (BovineHD0700027327); BTA 7: 111,481,071 bp (BovineHD0700032536); BTA 14: 1,463,676 bp (Hapmap30381-BTC-005750); BTA 16: 30,440,171 bp (BovineHD1600008636); BTA 20: 39,761,822 bp (ARS-BFGL-NGS-73590); and BTA 28: 45,702,356 bp (BovineHD2800013250) (Table 2). The Manhattan plots for the concentration of albumin in colostrum and serum are shown in Figure 3.

### 3.3. Candidate Genes and Functional Analysis

After mapping to the bovine genome assembly UMD3.1.1, we found 42 genes within the region that were 1 Mbp of up/downstream of the significant SNPs for the colostrum albumin concentration, including 36 protein-coding genes, 1 miRNA genes and 5 pseudogenes (Appendix A). By performing GO terms and KEGG pathways analysis, RUNX family transcription factor 1 (*RUNX1*), carbonyl reductase 1 (*CBR1*) and OTU deubiquitinase with linear linkage specificity (*OTULIN*) were observed to be involved in the positive regulation of interleukin-2 production, oxidation-reduction process, negative regulation of nuclear factor (NF)-kappaB transcription factor activity and negative regulation of inflammatory response (Appendix A). Hence, *RUNX1* (BTA 1: ~148.73 Mbp)*, CBR1* (BTA 1: ~150.06 Mbp) and *OTULIN* (BTA 20: ~58.58 Mbp) were eventually selected as the candidate genes for the colostrum albumin concentration (Table 3).

A total of 206 genes were obtained 1 Mbp from the significant SNPs for the serum albumin concentration, containing 155 protein-coding genes, 5 miRNA genes and 46 pseudogenes (Appendix A). Of these, the GO and KEGG results showed that 24 genes participated in the albumin-related terms and pathways, such as the oxidation–reduction process, regulation of inflammatory and immune response, and NF-kappa B and MAPK signaling pathway (Appendix A). After comparing the physical positions of these 206 genes with the positions of the released QTLs for inflammation and immune capacity in cattle (27 August 2020, Cattle QTLdb), 138 genes were found located in the interval of these QTLs. Combining the results of the GO and KEGG with the known QTL data, 12 functional genes were identified as promising candidates related to the serum albumin concentration (Table 3), including cyclin-dependent kinase 6 (*CDK6*, BTA 4: ~9.92 Mbp), SHANK-associated RH domain interactor (*SHARPIN*, BTA 14: ~1.93 Mbp), cytochrome c1 (*CYC1*, BTA 14: ~1.93 Mbp), exosome component 4 (*EXOSC4*, BTA 14: ~1.95 Mbp), poly (ADP-ribose) polymerase family member 10 (*PARP10*, BTA 14: ~2.03 Mbp), nuclear receptor binding protein 2 (*NRBP2,* BTA 14: ~2.16 Mbp), GDP-L-fucose synthase (*GFUS*, BTA 14: ~2.29 Mbp), pyrroline-5-carboxylate reductase 3 (*PYCR3,* BTA 14: ~2.31 Mbp), eukaryotic translation elongation factor 1 delta (*EEF1D*, BTA 14: ~2.32 Mbp), gasdermin D (*GSDMD*, BTA 14: ~2.34 Mbp), pyrroline-5-carboxylate reductase 2 (*PYCR2*, BTA 16: ~29.70 Mbp) and C-X-C motif chemokine ligand 12 (*CXCL12*, BTA 28: ~45.41 Mbp).

## 4. Discussion

In this study, we identified nine and seven genome-wide significant SNPs (FDR at 1%) associated with the albumin concentration in colostrum and serum in Chinese Holstein, respectively, using GWASs. For the significant SNPs captured in these two traits, no common genomic regions were shared. Hence, we calculated the correlations between these two traits using GCTA software and the phenotypic and genetic correlations were 0.011 and 0.0996, respectively, suggesting a weak correlation between the colostrum and serum albumin concentration. This is likely the reason that no common genomic regions were identified in the GWAS for these two traits. Besides, albumin is produced by hepatocytes and the majority of the protein is immediately released into the blood circulation [12] and only small amounts of albumin in the blood enters the milk by tight junction [8]. These two different sources and transport mechanisms of albumin in colostrum and serum might have been the cause of the weak phenotypic and genetic correlations between these two traits.

Population stratification and family structure can cause a number of false positive results in GWASs [52]. After a proper correction, the λ value should be close to 1 [53]. In the present study, the inflation factor (λ) was 0.983 and 1.004 for the concentration of colostrum and serum albumin, respectively, indicating that the population stratification was successfully corrected by the appropriate model [51]. 

In this study, the MAF of each significant SNP ranged from 0.107 to 0.463 for the colostrum and serum albumin concentrations. This implied these SNPs could be used for marker-assisted selection or a genomic selection program through selecting the advantageous alleles with positive effects to accelerate the albumin concentration in colostrum and serum in dairy cattle, thereby increasing the disease-resistance ability of calves. The contribution to genetic variance of the SNPs were relatively high with a range of 2.55–4.091% for the colostrum and serum albumin concentrations. Previous studies indicated that the SNPs identified by GWASs generally explain only a small fraction of the heritability, while the SNP effects might be magnified when applying the mixed-model association (MLMA) methods by GCTA [44,54,55]. Due to the advantages of the MLMA method in GCTA, like the prevention of false positive associations and an increase in power obtained through the application of a correction that is specific to a specific structure, MLMA is still a popular method for complex traits in GWASs [54].

After the functional analysis of genes in the regions within a 1 Mbp distance of the significant SNPs, 3 and 12 genes were identified as promising candidates for the concentration of colostrum and serum albumin in dairy cattle, respectively. Of these, *CBR1* and *CYC1* were reported to be associated with the oxidation–reduction process. *CBR1* encodes the protein belonging to the short-chain dehydrogenases/reductases (SDR) family that acts as an NADP-dependent oxidoreductase in the oxidation–reduction process [56]. The *CYC1* gene encodes a subunit of the cytochrome bc1 complex whose catalytic activity is required for the release of pro-apoptotic factors from mitochondria and the execution of the subsequent apoptotic steps [57,58,59]. It is well known that anti-oxidation is one of the most important properties of albumin. Remarkably, the oxidation-reduction process is the crucial part of an anti-inflammatory response, which demonstrates that the albumin in colostrum and serum is essential in anti-inflammation. 

The other set of candidate genes, i.e., *RUNX1, OTULIN, CDK6*, *SHARPIN*, *NRBP2*, *PYCR3*, *GSDMD*, *PYCR2* and *CXCL12*, were involved in immunity and inflammation. It has been proven that the mutations of *RUNX1* initiate the hyperactivation of inflammatory and innate immunity, including the IL-6, TLR, NF-kappaB, IFN and TREM1 signaling pathways [60]. *OTULIN* is critical for restraining life-threatening spontaneous inflammation, maintaining immune homeostasis and activating NF-kappaB to promote the secretion of pro-inflammatory cytokines and restricts bacterial proliferation in infection [61,62]. *CDK6* is required for the expression of inflammatory genes and is a critical regulator in the NF-kappaB signaling pathway, as well as contributing to cytokine production while inhibiting apoptosis [63,64]. *SHARPIN* regulates TLR3-mediated innate immunity, auto inflammation and the development of immunodeficiency [65,66]. The nuclear receptor binding protein, NRBP2, fights the infection of intracellular pathogens by regulating autophagy in the innate immune response [67]. *PYCR2* and *PYCR3* encode a protein that belongs to the pyrroline-5-carboxylate reductase family of enzymes that responds to genotoxic, inflammatory, nutrient and oxidative stress [68,69]. *GSDMD* encodes a member of the adermin family of pore-forming proteins implicated in the immune response [70], which controls the release of the proinflammatory cytokines IL-1ß, IL-18 and pyroptotic cell death, and drive the inflammation in septic shock as well as the autoimmune diseases [71,72]. *CXCL12* encodes a stromal cell-derived alpha chemokine member of the intercrine family and is involved in many diverse cellular functions, such as immune surveillance, inflammation response and tissue homeostasis [73,74,75]. Generally, all of these genes played vital roles in the inflammation and immune-related process, which indicated the potentially important roles of albumin in colostrum and serum in resistance to infectious diseases.

Resistance to viral infection is also a crucial capacity for the health and survival of calves. Herein, the candidate genes *PARP10*, *EXOSC4, GFUS* and *EEF1D*, involved in various viral infections, were identified. *PARP10,* a member of the poly (ADP-ribose) polymerases (PARPs) family, is related to immunity, metabolism, apoptosis and DNA damage repair [76], and could alter the cell cycle to inhibit virus replication during the process of avian influenza virus infection [77]. *EXOSC4* participated in the regulation of anti-viral responses to decrease human papillomavirus infectivity of keratinocytes [78]. *GFUS* activated the immune-network to enhance the regulation of apoptosis, T cell homeostasis, neutrophil-mediated immunity, neutrophil chemotaxis, interleukin-8 production, inflammatory response, immune response, B-cell activation and MAPK activity activation during hepatitis C virus infection [79]. During human immunodeficiency virus 1 (HIV-1) infection, *EEF1D* interacted with HIV-1 transcription protein, resulting in the inhibition of the translation of host cell proteins but in an increase in the translation of viral proteins [80]. These studies suggested the importance of albumin together with these candidate genes in fighting infections caused by viruses.

Moreover, from a breeding perspective, the concentration of albumin in serum or milk is influenced by pathological and genetic factors, which highlights the possibility of albumin as a new trait to improve dairy cattle disease resistance. Nowadays, with the comprehensive implication of genomic selection in dairy cattle breeding, where high-density SNP chips, such as Illumina 50 K and GeneSeek 150 K chips, are widely used, most of SNPs are collected from the current SNP database and almost evenly distributed across the whole genome. Therefore, the significant SNPs and genes associated with albumin concentration could be put into such chips to improve the health and resistance to pathogens of newborns in dairy cattle.

## 5. Conclusions

In summary, our GWAS detected nine and seven genome-wide significant SNPs associated with the concentration of albumin in colostrum and serum, respectively. By integrated analysis of the biological functions of the genes that contain or close to (±1 Mbp) such significant SNPs and the known QTLs for inflammation and immunity capacity, *RUNX1*, *CBR1* and *OTULIN* were selected as the candidate genes for the albumin concentration in colostrum. Meanwhile, 12 promising candidate genes were suggested for the albumin concentration in colostrum and serum of dairy cattle, including *CDK6*, *SHARPIN*, *CYC1*, *EXOSC4*, *PARP10*, *NRBP2*, *GFUS*, *PYCR3*, *EEF1D*, *GSDMD*, *PYCR2* and *CXCL12*. Our results provided a genetic view on the regulation of albumin and fundamental information for the genetic improvement program on health and resistance traits in dairy cattle.

## Figures and Tables

**Figure 1 animals-10-02211-f001:**
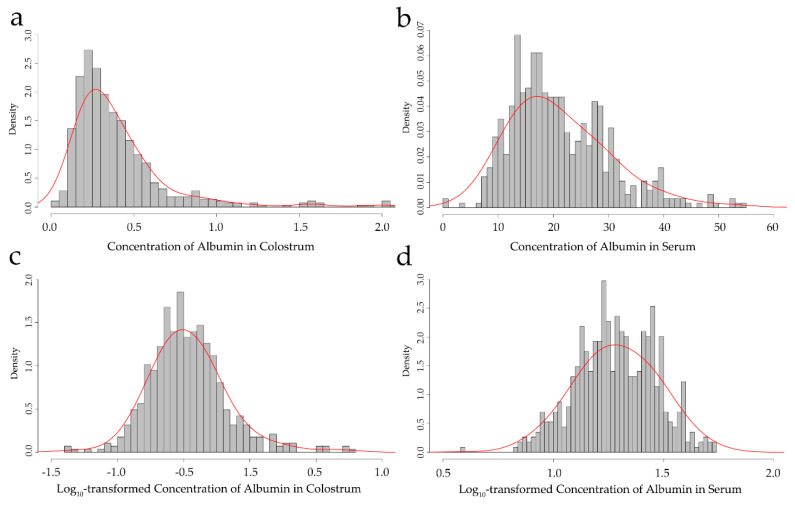
Frequency distribution of the concentration of albumin in colostrum and serum: (**a**,**b**) show the original colostrum and serum albumin concentrations; (**c**,**d**) show the log_10_-transformed colostrum and serum albumin concentrations.

**Figure 2 animals-10-02211-f002:**
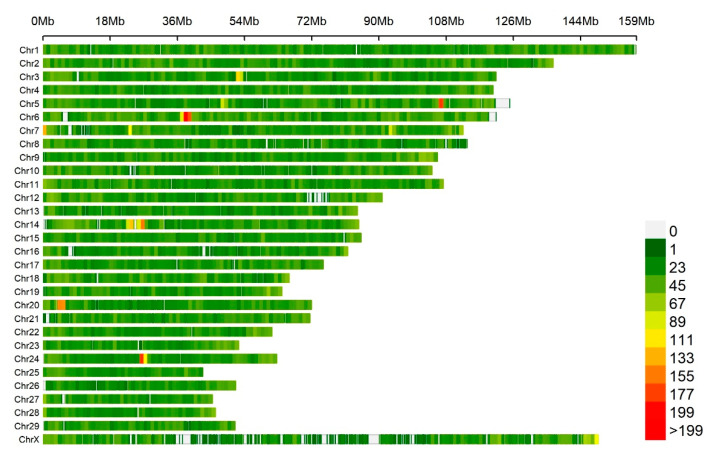
The SNPs’ density distribution on 29 autosomes and the X-chromosome of the bovine genome. The horizontal axis (*X*-axis) shows the chromosome length (Mbp). SNP density was calculated per 1 Mbp window. Different colors represent different SNP density levels.

**Figure 3 animals-10-02211-f003:**
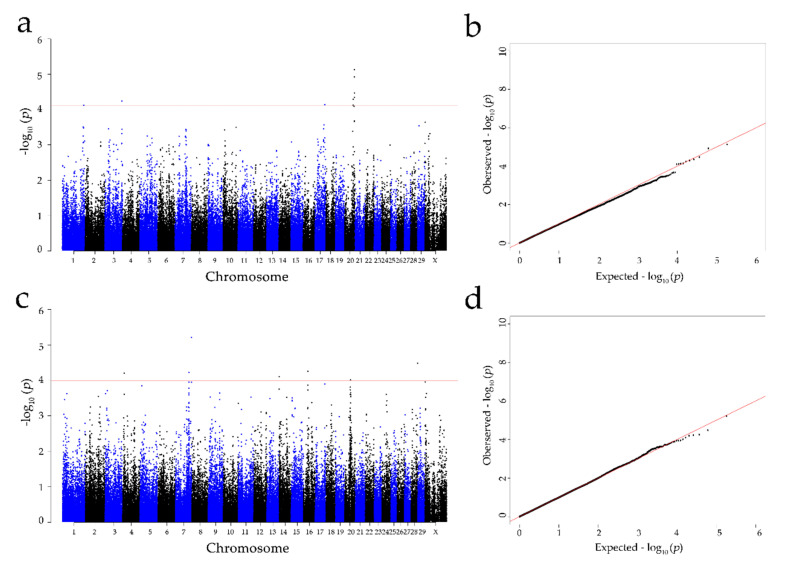
Manhattan and Q-Q plots of the observed *p*-values for the concentration of albumin in the colostrum and serum: (**a**,**b**) indicate the albumin concentration in colostrum; (**c**,**d**) indicate the albumin concentration in serum. The Manhattan plots present the −log_10_ values (*p*-values) for the genome-wide SNPs (*y*-axis) plotted against their respective positions on each chromosome (*x*-axis); the horizontal red in the Manhattan plots present the thresholds with an FDR rate of 1% for albumin in colostrum (7.85 × 10^5^) and serum (1.01 × 10^5^). The Q-Q plots show the observed −log_10_-transformed *p*-values (*y*-axis) and the expected −log_10_-transformed *p*-values (*x*-axis).

**Table 1 animals-10-02211-t001:** Descriptive statistics for the concentration of albumin in colostrum and serum; *n* = 572.

Traits ^a^	Arithmetic Mean (mg/mL)	SD ^b^	Minimum	Maximum	CV ^c^	Skewness	Kurtosis
col	0.454	0.555	0.040	6.180	1.224	6.065	46.956
ser	21.061	8.883	0.220	54.500	0.422	0.909	0.957
col-log_10_	−0.466	0.291	−1.390	0.790	−0.625	0.737	2.213
ser-log_10_	1.282	0.213	−0.660	1.740	0.166	−2.595	21.913

^a^ col and ser = the concentration of albumin in colostrum and serum, respectively; log_10_ = log_10_-transformed phenotypes; ^b^ standard deviation; ^c^ coefficient of variation.

**Table 2 animals-10-02211-t002:** The significant SNPs of the genome-wide association studies for the concentration of albumin in colostrum and serum.

Traits ^a^	Chr ^b^	SNP	Position (bp) on UMD 3.1	Major/Minor Allele	MAF ^c^	SNP Effect	SE ^d^	CGV(%) ^e^	FDR-Corrected *p*-Value
col	1	BovineHD0100043239	149,246,858	C/A	0.156	0.176	0.044	2.550	9.765 × 10^5^
col	3	ARS-BFGL-NGS-75987	118,061,120	C/A	0.127	0.207	0.052	3.188	7.258 × 10^5^
col	17	ARS-BFGL-NGS-66134	68,421,115	G/A	0.379	0.137	0.035	2.869	9.248 × 10^5^
col	20	BovineHD2000016546	59,137,600	G/A	0.420	−0.136	0.034	2.869	9.506 × 10^5^
col	20	BovineHD2000016866	60,034,418	A/G	0.351	0.142	0.035	2.869	6.600 × 10^5^
col	20	BovineHD2000019816	68,269,626	A/G	0.179	0.191	0.043	3.506	9.442 × 10^5^
col	20	BTB-00798071	68,605,103	C/A	0.107	0.243	0.055	3.506	1.513 × 10^5^
col	20	ARS-BFGL-NGS-75636	69,893,541	G/A	0.146	0.199	0.048	3.188	4.329 × 10^5^
col	20	ARS-BFGL-NGS-114933	69,916,426	A/G	0.149	0.196	0.048	3.188	5.724 × 10^5^
ser	4	Hapmap39425-BTA-70290	10,737,673	A/C	0.414	2.241	0.560	3.062	6.282 × 10^5^
ser	7	BovineHD0700027327	93,597,405	G/A	0.396	2.450	0.610	3.607	5.881 × 10^5^
ser	7	BovineHD0700032536	11,1481,071	G/A	0.163	3.451	0.763	4.091	6.032 × 10^5^
ser	14	Hapmap30381-BTC-005750	1,463,676	G/A	0.328	2.432	0.616	3.273	7.814 × 10^5^
ser	16	BovineHD1600008636	30,440,171	G/A	0.459	−2.349	0.583	3.443	5.558 × 10^5^
ser	20	ARS-BFGL-NGS-73590	39,761,822	A/G	0.407	2.328	0.597	3.289	9.636 × 10^5^
ser	28	BovineHD2800013250	45,702,356	G/A	0.463	−2.383	0.574	3.547	3.252 × 10^5^

^a^ col and ser = the concentration of albumin in colostrum and serum, respectively; ^b^ cow chromosome; ^c^ minor allele frequency; ^d^ standard error of the SNP effect; ^e^ contribution to genetic variance.

**Table 3 animals-10-02211-t003:** Candidate genes 1 Mbp from the significant SNPs identified in the genome-wide association studies for the albumin concentration in colostrum and serum.

Gene ID	Chr ^a^	Gene Name	Gene Start (bp) ^b^	Gene End (bp) ^b^	Traits ^c^
ENSBTAG00000004742	1	*RUNX1*	148,678,710	148,773,781	col
ENSBTAG00000023384	1	*CBR1*	150,054,221	150,064,637	col
ENSBTAG00000003186	20	*OTULIN*	58,563,064	58,596,022	col
ENSBTAG00000044023	4	*CDK6*	9,791,798	10,039,688	ser
ENSBTAG00000012235	14	*SHARPIN*	1,925,026	1,929,354	ser
ENSBTAG00000012232	14	*CYC1*	1,930,183	1,932,580	ser
ENSBTAG00000014607	14	*EXOSC4*	1,947,198	1,949,074	ser
ENSBTAG00000009677	14	*PARP10*	2,024,591	2,031,476	ser
ENSBTAG00000008079	14	*NRBP2*	2,154,132	2,159,657	ser
ENSBTAG00000034691	14	*GFUS*	2,288,555	2,293,395	ser
ENSBTAG00000016810	14	*PYCR3*	2,301,587	2,309,099	ser
ENSBTAG00000014643	14	*EEF1D*	2,314,039	2,326,727	ser
ENSBTAG00000021474	14	*GSDMD*	2,341,282	2,347,798	ser
ENSBTAG00000005835	16	*PYCR2*	29,695,733	2,9699,696	ser
ENSBTAG00000005077	28	*CXCL12*	45,410,676	45,418,794	ser

^a^ Cow chromosome number; ^b^ the position of gene was based on the UMD 3.1.1 assembly; ^c^ col and ser = the concentration of albumin in colostrum and serum, respectively.

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
