# Peer review of "Genome-Wide Association Studies for the Concentration of Albumin in Colostrum and Serum in Chinese Holstein"

_animals, 2020, doi:10.3390/ani10122211_

Round 1
Reviewer 1 Report
This is a very good paper, well done.
L38: Alphabetical order
Author Response
Dear Reviewer:
Thank you very much for your positive comments and constructive suggestion on the manuscript. We have made the revision as you suggested.
L38: Alphabetical order
Answer: Done as requested (Line 38).
Finally, thank you again for the valuable suggestion. If you have any question on this manuscript, please feel free to contact me. Thank you again for your time and favorable consideration.
Sincerely,
Dongxiao Sun, PhD, Professor
Tel/Fax: +86-10-62734653
Email: [email protected]
Reviewer 2 Report
The study by Shan Li et al. describes a GWAS analysis on albumin concertation measured either in colostrum or blood serum on 613 dairy cows. The overall objective is to identify genomic regions that might be regulating albumin concertation in dairy cattle; information that could be further incorporated in breeding programs via markers assisted selection. Indeed, several diseases exist in dairy cattle related to postpartum condition of the cows, while newborn calves’ mortality during the first days after birth (especially due to E.coli) is a common problem in dairy farming. Colostrum feeding to newborn calves immediately after birth and within a time-frame of 5-6h is a common practise and has been associated with an increasing survivability of young calves. Colostrum is rich in albumin and immunoglobulins and provides newborn calves with the necessary antibodies against several pathogens.
Although the study is of general interest, before further consideration, major revisions are needed to show the real significance and impact of the study. Moreover, there are several points that need to be addressed and do not justify publication at present.
Major points that should be addressed are:
- Albumin concertation in colostrum as well as the ability of new-born calves to absorb proteins drastically decrease within few hours after calving. For this reason, it is a common practise in dairy farms to feed new-born calves with colostrum immediately after calving or within 5-6 hours at maximum. However, samples of this study were collected within 24 hours from calving. This also means that cows were not milked for ~1 day after calving (!) which is quite strange. Why a period of 24h after calving was decided for sampling?
- Milk and blood samples per cow were collected exactly at the same time point to measure the albumin? If not, for e.g., if in a given cow the colostrum was collected 1 h after calving while blood sample was taken 23h later, could this difference in time-point have an impact in the analysis?
- Why albumin was measured both in colostrum and serum? This is not clear in the manuscript. Do we expect that the 2 measures are actually two different traits? Moreover, why an ELISA kit was used? What is the accuracy of the ELISA kit in measuring albumin?
- Related to the point above, what is the phenotypic and genetic correlation between the 2 albumin concentrations (colostrum/blood)? Given the GWAS results, the traits seem to be genetically uncorrelated, since they do not share any common genomic regions identified by GWAS, not even a common chromosome (with the exception of X-chromosome; but also in this case the genomic regions clearly differ).
- Why the albumin concertation is not following a normal distribution? Please provide with a plot of the distribution and report skewness, kurtosis and coefficient of variation for each trait (measured or log-transformed) in the manuscript.
- It is not clear in the manuscript if albumin is related to healthier cows or calves. I assume that the general problem to be addressed is the increased mortality of new-borns in dairy cattle rather postpartum diseases of cows. This is rather confusing in the manuscript.
- Please provide with more details regarding the sampling period. It is also not clear if 2 to 160 cows were sampled per herd or this is the general number of cows/herd.
- Regarding imputation and GWAS, I have the concern that by including 32 imputed cows you have created more problems than solving. What is the point to increase the sample size in a GWAS from 581 to 613? Do you expect a drastic increase in the power of GWAS to detect significant SNP by adding 32 samples (5% increase of sample size)? On the other hand, I think that with 32 imputed cows you have increased the error in the genotypes. Imputation is based, in general, in a small set of core bulls representative of the whole population and genotyped with HD. In this case, imputation can provide with reasonable results. Up to what extend was the relationship between the LD and HD genotyped cows? Moreover, I would be really interested in the results of a GWAS by using only the 581 cows genotyped in HD compared with the results including the 32 cows imputed to HD density. In this way you decrease errors in the genotypes while keeping the same power to detect significant SNP. It sounds also strange that after QC no samples were removed, although the threshold of missingness was set to 0.90; I could argue that a value of 0.95 is a more reasonable threshold to be used. Overall, my suggestion is to re-run the GWAS including only the 583 HD cows, with a threshold of missingness per sample of 0.95 and perhaps with an FDR at 5%. FDR at 1% might be too strict for a sample size of ~600 cows, so you might lose some important information. You could compare then the results with those reported in the manuscript.
- What is the additive genetic variance explained by each significant SNP or 1Mbp SNP windows? This information is important and will add credits to your work.
- In my opinion the presentation of the results is currently not sufficient.
- The Discussion is not serving to its purpose. The Discussion section should be used to organize the results into some context, draw conclusions, and show their significance and implications. Do not simply “throw” in the text SNP/gene names and the metabolic pathways they are involved in based on the literature. The results have to be summarized in a way that they are directly linked to the scientific question of your research. For e.g., how is linked the inflammatory responses to cigarette smoke of MAP2K3 with the mortality rate of newborn calves (L220-221)?
Minor comments:
The text contains many syntax errors. I listed a few of these errors below, but this list represents a random subselection only. I suggest all authors to proofread the text prior to resubmission.
- L14 report some examples of diseases you are referring to, both for cows and newborn calves; replace “The clinical” with clinical; replace “postpartal” with “postpartum” (here and throughout the manuscript); replace “the newborn calves” with “newborn calves”.
- L15 replace “resulted” with “result”.
- L17 The term albumin refers to a group of proteins.
- L18 replace “fight” with “against”.
- L20 GWAS was not abbreviated before, use the full name
- L26 “…concentration of albumin,” where? Please, be specific.
- L29 SNPs was not abbreviated before, use the full name.
- L42 replace “objectives” with “objective”; you could also use a more recent publication regarding breeding goals in dairy cattle: Miglior, F., Fleming, A., Malchiodi, F., Brito, L.F., Martin, P., Baes, C.F. A. 2017. 100-Year Review: Identification and genetic selection of economically important traits in dairy cattle. J. Dairy Sci. 100:10251-10271.
- L43 report some examples of diseases you are referring to, both for cows and newborn calves.
- L50 replace “The previous” with “Previous”.
- L51-52 provide with species specific literature, e.g. cows [citation], sheep [citation] and goats [citation].
- L54 replace “the concentration” with “concentration”.
- L55 replace “found was high” with “found high”; replace “the mature” with “mature”.
- L56-58 please, rephrase the whole sentence.
- L61-62 This statement is wrong. Is there any support in the literature about the polygenic character of albumin? This is a hypothesis. This is also in contrast to your findings where only 15 genes were reported, indicating an oligogenic trait.
- L62 replace “Precious” with “Previous”.
- L63 replace “Similarly, the heritability” with “Heritability”.
- L67-70 please split the citation by group of traits, e.g. milk yield traits [citation], health [citation], reproduction [citation] and conformation [citation]. You help the reader to focus and you avoid long citation lists.
- L70-71 please rephrase the sentence. For e.g., “However, there is a limited number of studies that have investigated…”.
- L80-81 please remove the average values of parity and cows/sire. What was the age of the cows?
- L87-89 please rephrase. You could simply say that concentrations were log-transformed to follow a normal distribution. Please also indicate the base of the logarithm.
- L100-101 rephrase “were included in the further association analysis” with “were kept for further analysis” or “were further analyzed in GWAS”.
- L105 & L109 the m is a matrix of genotypes, should be in capital.
- L110 G refers to a matrix, so it should be in bold; do not abbreviate GRM since the term appears only once in the manuscript. Do you also used SNP from the X-CHR to construct the GRM? There are several methods to construct a GRM, please indicate the method used.
- L112 remove one extra 0 in the parenthesis describing the residual distribution; explain that I is a diagonal matrix; I is a matrix and should be in bold.
- L113 replace “90.2” with “v1.90.2”.
- L118 replace “m” in the equation with another letter to avoid confusion with the “m” reported in the equation on L105.
- L122-123 citation is missing for genabel, R 3.6.0 and qqman.
- L123 delete “in R 3.6.0”, it is already mentioned in the previous sentence.
- L141 delete “total”.
- L142 replace “their distribution was” with “their distribution on the genome is”.
- Table 1 please report skewness, kurtosis and coefficient of variation for each trait.
- L143 remove the original and corrected from the title and keep only the concentration of the albumin in colostrum and the serum. Below Table 1 you provide with explanation.
- L 145-146 replace “log, corrected ... transformation” with “log-transformed”.
- In Figure 1 and throughout the manuscript please replace Chr30 with X-chromosome.
- L148 replace “on every chromosome” with “on 29 autosomes and the X-chromosome (Chr30)”.
- L149 replace “showed” with “shows”
- L155 replace “FDR p-value < 0.01” with “FDR at 1%”.
- L156 replace “locating” with “located”; Bos taurus in italic
- L156-159 please report SNP name and position on the chromosome. The number of SNP/CHR does not provide with useful information to the reader.
- Table 2 replace “Triats” with “Traits”; Remove “_A” since it is common in all rows. It is explanatory to report col and ser to indicate the different concertation of albumin; Also include comma separator in the column referring to the position of the SNP; Please replace Chr30 with X-chromosome
- L164 replace “standard error” with “standard error of the SNP effect”.
- Figure 2 is of poor quality, please replace.
- L173-191 This part needs re-writing. It would be much easier to the reader if you split the results by trait. Moreover, please, for each gene report position on the genome (CHR and approximate position in Mbp).
- L192 replace “The information of …association…” with “Candidate genes with 1Mbp from the significant SNP identified in GWAS analyses of albumin …” .
- L202 replace “resultsin” with “results in”.
- L211 replace “function” with “functional”.
- L219 which previous study are you referring too?
- L221 how is the MAP2K3 and cigarette smoke related to the mortality of newborn calves?
- L226 SHMT in italic
- L254 GSDMD in italic.
- L259 CXCL12 in italic.
- L263 replace “So for” with “So far”. In general, this sentence needs re-writing. Please, indicate the values of the correlations mentioned.
- L273 PYCR3 in italic
Author Response
Dear Reviewer:
Thank you very much for your positive comments and constructive suggestions on the manuscript. We have made the essential revisions point by point as you suggested.
- Albumin concertation in colostrum as well as the ability of new-born calves to absorb proteins drastically decrease within few hours after calving. For this reason, it is a common practice in dairy farms to feed new-born calves with colostrum immediately after calving or within 5-6 hours at maximum. However, samples of this study were collected within 24 hours from calving. This also means that cows were not milked for ~1 day after calving (!) which is quite strange. Why a period of 24h after calving was decided for sampling?
Answer: Thanks so much for the Review’s comment and suggestion. Actually, in our study, almost 80% colostrum samples and serum were collected within 6 hours from calving in which the majority were collected immediately. As for the minority samples collected more than 6 hours, these cows calved at midnight when workers and veterinarians were sleeping or rest so it can't avoid collecting the colostrum in the next day, resulting in a few samples were obtained more than 6 hours from calving. Based on this, when we re-run the GWAS with the 573 cows with 150K chips, we added the fix effect of collecting time (level 1:0-6 hour, level 2: 6-12 hour, level 3: 12-18 hour, level 4: 12-24 hour) in the mixed linear model to correct the difference of albumin concentration across different sampling time (Lines 74-76 and 100).
- Milk and blood samples per cow were collected exactly at the same time point to measure the albumin? If not, for e.g., if in a given cow the colostrum was collected 1 h after calving while blood sample was taken 23h later, could this difference in time-point have an impact in the analysis?
Answer: Thanks so much for the Review’s comment and suggestion. Yes, in this study, the veterinarians collected the milk and blood samples per cow at the same time. We separated the serum by centrifugation after clotting and stored the colostrum and serum samples at -20℃ for further unified measurement of albumin concentration.
- Why albumin was measured both in colostrum and serum? This is not clear in the manuscript. Do we expect that the 2 measures are actually two different traits? Moreover, why an ELISA kit was used? What is the accuracy of the ELISA kit in measuring albumin?
Answer: Thanks so much for the Review’s comment and suggestion. It is well known that the colostrum albumin is important for newborn calves. Actually, previous studies proved that colostrum albumin was derived from the blood plasma through the epithelial tight junction (Jordan & Morgan 1967; Schanbacher & Smith 1975; Lascelles 1979; de Wit 1998). Hence, we analyzed these 2 two related traits in this study. In addition, the ELISA kit can directly detect the absolute content of albumin in milk and serum and this is currently the most commonly recognized method for measurement of albumin concentration. The ELISA kit used in this study can detect albumin concentration ranged from 0.69 to 500ng/ml and our minimum concentration was 40ng/ml, ensuring the accuracy of measurement.
- Related to the point above, what is the phenotypic and genetic correlation between the 2 albumin concentrations (colostrum/blood)? Given the GWAS results, the traits seem to be genetically uncorrelated, since they do not share any common genomic regions identified by GWAS, not even a common chromosome (with the exception of X-chromosome; but also in this case the genomic regions clearly differ).
Answer: Thanks so much for the Review’s comment and suggestion. We totally agreed with your comment that the 2 albumin concentrations in colostrum and blood might show some phenotypic and genetic correlations. As suggested, based on our population and phenotype data, we calculated the genetic correlation between these 2 traits by GCTA software and the phenotypic and genetic correlation were 0.011 and 0.0996, suggesting the weak correlation between the colostrum and serum albumin concentration. This is likely the reason that no common genomic regions identified by GWAS for these 2 traits. Albumin is produced by hepatocytes and the majority of the protein is immediately released into the blood circulation and only small amounts of albumin in the blood enters the milk by tight junction (Jordan & Morgan 1967; Rozga et al. 2013). The weak phenotypic and genetic correlations between these two traits may be due to these two different source and transport mechanism of albumin in colostrum and serum. We have added these contents in the Discussion section (Lines 210-219). If you suggest to add the content about correlation analysis in the Materials and Methods and Results section, we would like to follow your suggestion in the revised manuscript.
- Why the albumin concertation is not following a normal distribution? Please provide with a plot of the distribution and report skewness, kurtosis and coefficient of variation for each trait (measured or log-transformed) in the manuscript.
Answer: Thanks so much for the Review’s comment and suggestion. In fact, the skewed distributions of some traits are very common in many previous studies, such as the immunoglobulins concentrations in colostrum and serum in our previous GWASs in the same Chinese Holstein population (submitted, Shan Lin et al. 2020), immunoglobulins in blood and milk for GWASs in Canadian and Dutch Holstein cattle (de Klerk et al. 2018; Cordero-Solorzano et al. 2019). In this study, concentrations of albumin in colostrum and serum were log10-transformed to follow a normal distribution for further GWAS. Moreover, we have already provided the plots of the distribution and report skewness, kurtosis and coefficient of variation for each trait (measured or log-transformed) in the manuscript (Figure1 and Table 1).
- It is not clear in the manuscript if albumin is related to healthier cows or calves. I assume that the general problem to be addressed is the increased mortality of new-borns in dairy cattle rather postpartum diseases of cows. This is rather confusing in the manuscript.
Answer: Thanks so much for the Review’s comment and suggestion. We quite agree with your opinion and have already revised our manuscript mainly in Abstract (Lines 14-17, 24), Introduction (Lines 42-46, 55-56, 70-71) and Discussion sections (Lines 284-291).
- Please provide with more details regarding the sampling period. It is also not clear if 2 to 160 cows were sampled per herd or this is the general number of cows/herd.
Answer: Done as requested. We have provided more details regarding the sampling period and number of cows of every herd as follows:
The blood, colostrum and hair follicle samples were collected in the first milking within 24h after calving from 572 Chinese Holstein cows (0-6h: 481 cows, 6-12h: 38 cows, 12-18h: 39 cows, 12-24h: 14 cows). All the cows were from 10 dairy farms in the Beijing Dairy Cattle Center and the Beijing Sunlon Livestock Development Company Limited (herd 1: 90 cows, herd 2: 5 cows, herd 3: 88 cows, herd 4: 92 cows, herd 5: 58 cows, herd 6: 47 cows, herd 7: 75 cows, herd 8: 56 cows, herd9: 19 cows, herd 10: 42 cows) (Lines 75-79).
- Regarding imputation and GWAS, I have the concern that by including 32 imputed cows you have created more problems than solving. What is the point to increase the sample size in a GWAS from 581 to 613? Do you expect a drastic increase in the power of GWAS to detect significant SNP by adding 32 samples (5% increase of sample size)? On the other hand, I think that with 32 imputed cows you have increased the error in the genotypes. Imputation is based, in general, in a small set of core bulls representative of the whole population and genotyped with HD. In this case, imputation can provide with reasonable results. Up to what extend was the relationship between the LD and HD genotyped cows? Moreover, I would be really interested in the results of a GWAS by using only the 581 cows genotyped in HD compared with the results including the 32 cows imputed to HD density. In this way you decrease errors in the genotypes while keeping the same power to detect significant SNP. It sounds also strange that after QC no samples were removed, although the threshold of missingness was set to 0.90; I could argue that a value of 0.95 is a more reasonable threshold to be used. Overall, my suggestion is to re-run the GWAS including only the 583 HD cows, with a threshold of missingness per sample of 0.95 and perhaps with an FDR at 5%. FDR at 1% might be too strict for a sample size of ~600 cows, so you might lose some important information. You could compare then the results with those reported in the manuscript.
Answer: Thanks so much for the Review’s comment and suggestion. We have re-run the GWASs including only the cows with 150k chip with a threshold of missingness per sample of 0.95, in which totally 563 samples with 91,620 SNPs were used. As a result, using the FDR at 5% as the threshold, we identified 28 and 53 significant SNPs for colostrum and serum albumin concentration, respectively. Considering the false positive rate, FDR at 1% was finally set as the threshold of significance. Consequently, 9 and 7 genome-wide significant SNPs were identified for colostrum and serum albumin concentration, respectively. Of these, compared with the results of the last GWAS, for colostrum albumin, 6 adjacent significant SNPs (BTA 20: 59137600~ 69916426 bp) were also identified or close to the significant SNP (BTA 20: 59137600 bp); For serum albumin, 3 SNPs (BTA 7: 93597405 and 111481071 bp, BTA 14: 1463676 bp) were the same as or near the significant SNPs identified last time. In addition, 3 and 4 additional SNPs were identified after re-running GWAS for colostrum and serum albumin, respectively. Correspondingly, Abstract (Lines 27, 31 and 34), Materials and Methods (Lines 87, 91 and 93), Results (Lines 130,133, 148-160 and 176-202), Discussion (Lines 209-283), Conclusion (Lines 293-301), Figure 1-4 and Table 1-3 were all revised.
- What is the additive genetic variance explained by each significant SNP or 1Mbp SNP windows? This information is important and will add credits to your work. In my opinion the presentation of the results is currently not sufficient.
Answer: Thanks so much for the Review’s comment and suggestion. We have calculated the additive genetic variance explained by each significant SNP which was added in the Table 2.
- The Discussion is not serving to its purpose. The Discussion section should be used to organize the results into some context, draw conclusions, and show their significance and implications. Do not simply “throw” in the text SNP/gene names and the metabolic pathways they are involved in based on the literature. The results have to be summarized in a way that they are directly linked to the scientific question of your research. For e.g., how is linked the inflammatory responses to cigarette smoke ofMAP2K3 with the mortality rate of newborn calves (L220-221)?
Answer: Thanks so much for the Review’s comment and suggestion. We have rewritten the discussion part as request (Lines 209-291).
Minor comments:
The text contains many syntax errors. I listed a few of these errors below, but this list represents a random subselection only. I suggest all authors to proofread the text prior to resubmission.
L14 report some examples of diseases you are referring to, both for cows and newborn calves; replace “The clinical” with clinical; replace “postpartal” with “postpartum” (here and throughout the manuscript); replace “the newborn calves” with “newborn calves”.
Answer: Done as requested (Line 14).
L15 replace “resulted” with “result”.
Answer: Done as requested (Line 14).
L17 The term albumin refers to a group of proteins.
Answer: Done as requested (Line 17).
L18 replace “fight” with “against”.
Answer: Done as requested (Line 18).
L20 GWAS was not abbreviated before, use the full name
Answer: Done as requested (Line 21).
L26 “…concentration of albumin,” where? Please, be specific.
Answer: Done as requested (Line 26).
L29 SNPs was not abbreviated before, use the full name.
Answer: Done as requested (Line 29).
L42 replace “objectives” with “objective”; you could also use a more recent publication regarding breeding goals in dairy cattle: Miglior, F., Fleming, A., Malchiodi, F., Brito, L.F., Martin, P., Baes, C.F. A. 2017. 100-Year Review: Identification and genetic selection of economically important traits in dairy cattle. J. Dairy Sci. 100:10251-10271.
Answer: Done as requested (Line 42).
L43 report some examples of diseases you are referring to, both for cows and newborn calves.
Answer: Done as requested (Line 43).
L50 replace “The previous” with “Previous”.
Answer: Done as requested (Line 51).
L51-52 provide with species specific literature, e.g. cows [citation], sheep [citation] and goats [citation].
Answer: Done as requested (Line 53).
L54 replace “the concentration” with “concentration”.
Answer: Done as requested (Line 50).
L55 replace “found was high” with “found high”; replace “the mature” with “mature”.
Answer: Done as requested (Line 50).
L56-58 please, rephrase the whole sentence.
Answer: This sentence has been deleted.
L61-62 This statement is wrong. Is there any support in the literature about the polygenic character of albumin? This is a hypothesis. This is also in contrast to your findings where only 15 genes were reported, indicating an oligogenic trait.
Answer: Thanks so much for the Review’s comment and suggestion. We totally agreed with your opinion and have re-written this sentence (Line 57).
L62 replace “Precious” with “Previous”.
Answer: Done as requested (Line 57).
L63 replace “Similarly, the heritability” with “Heritability”.
Answer: Done as requested (Line 59).
L67-70 please split the citation by group of traits, e.g. milk yield traits [citation], health [citation], reproduction [citation] and conformation [citation]. You help the reader to focus and you avoid long citation lists.
Answer: Done as requested (Lines 65-67).
L70-71 please rephrase the sentence. For e.g., “However, there is a limited number of studies that have investigated…”.
Answer: Done as requested (Line 67).
L80-81 please remove the average values of parity and cows/sire. What was the age of the cows?
Answer: Done as requested (Line 79). Cows were 23-72 months age at this time of calving (Line 80).
L87-89 please rephrase. You could simply say that concentrations were log-transformed to follow a normal distribution. Please also indicate the base of the logarithm.
Answer: Done as requested (Line 85).
L100-101 rephrase “were included in the further association analysis” with “were kept for further analysis” or “were further analyzed in GWAS”.
Answer: Done as requested (Line 93).
L105 & L109 the m is a matrix of genotypes, should be in capital.
Answer: Done as requested (Lines 92 and 102).
L110 G refers to a matrix, so it should be in bold; do not abbreviate GRM since the term appears only once in the manuscript. Do you also used SNP from the X-CHR to construct the GRM? There are several methods to construct a GRM, please indicate the method used.
Answer: We have replaced G in bold and added full name of GRM (Lines 103-104). We constructed GRM between pairs of individuals from SNPs of all 30 chromosome including X-CHR and save the lower triangle elements of the GRM to binary files by GCTA software (Lines 104 and 105).
L112 remove one extra 0 in the parenthesis describing the residual distribution; explain that I is a diagonal matrix; I is a matrix and should be in bold.
Answer: Done as requested (Line 105).
L113 replace “90.2” with “v1.90.2”.
Answer: Done as requested (Line 107).
L118 replace “m” in the equation with another letter to avoid confusion with the “m” reported in the equation on L105.
Answer: Done as requested (Line 112 and 114).
L122-123 citation is missing for genabel, R 3.6.0 and qqman.
Answer: Done as requested (Line 116).
L123 delete “in R 3.6.0”, it is already mentioned in the previous sentence.
Answer: Done as requested (Line 117).
L141 delete “total”.
Answer: Done as requested (Line 131).
L142 replace “their distribution was” with “their distribution on the genome is”.
Table 1 please report skewness, kurtosis and coefficient of variation for each trait.
Answer: Done as requested (Line 135).
L143 remove the original and corrected from the title and keep only the concentration of the albumin in colostrum and the serum. Below Table 1 you provide with explanation.
Answer: Done as requested (Line 141).
L 145-146 replace “log, corrected ... transformation” with “log-transformed”.
In Figure 1 and throughout the manuscript please replace Chr30 with X-chromosome.
Answer: Done as requested (Line 142). We have replace Chr30 with X-chromosome in Figure 2 and 3.
L148 replace “on every chromosome” with “on 29 autosomes and the X-chromosome (Chr30)”.
Answer: Done as requested (Line 145).
L149 replace “showed” with “shows”
Answer: Done as requested (Line 146).
L155 replace “FDR p-value < 0.01” with “FDR at 1%”.
Answer: Done as requested (Line 152).
L156 replace “locating” with “located”; Bos taurus in italic
Answer: Done as requested (Line 152).
L156-159 please report SNP name and position on the chromosome. The number of SNP/CHR does not provide with useful information to the reader.
Answer: Done as requested (Line 152-161).
Table 2 replace “Triats” with “Traits”; Remove “_A” since it is common in all rows. It is explanatory to report col and ser to indicate the different concertation of albumin; Also include comma separator in the column referring to the position of the SNP; Please replace Chr30 with X-chromosome
Answer: Done as requested (Table 2).
L164 replace “standard error” with “standard error of the SNP effect”.
Answer: Done as requested (166).
Figure 2 is of poor quality, please replace.
Answer: We have replaced it in high quality (Figure 3).
L173-191 This part needs re-writing. It would be much easier to the reader if you split the results by trait. Moreover, please, for each gene report position on the genome (CHR and approximate position in Mbp).
Answer: We have re-written this part as request (Lines 176-202).
L192 replace “The information of …association…” with “Candidate genes with 1Mbp from the significant SNP identified in GWAS analyses of albumin …” .
Answer: Done as requested (Lines 203-204).
L202 replace “resultsin” with “results in”.
Answer: Done as requested (Line 220).
L211 replace “function” with “functional”.
Answer: Done as requested (Line 225).
L219 which previous study are you referring too?
Answer: This part of the content has been deleted because of the result adjustment.
L221 how is the MAP2K3 and cigarette smoke related to the mortality of newborn calves?
Answer: We have re-written whole Discussion part and candidate gene MAP2K3 has been deleted because of the result adjustment.
L226 SHMT in italic
Answer: candidate gene SHMT has been deleted because of the result adjustment.
L254 GSDMD in italic.
Answer: Done as requested (Line 274).
L259 CXCL12 in italic.
Answer: Done as requested (Line 277).
L263 replace “So for” with “So far”. In general, this sentence needs re-writing. Please, indicate the values of the correlations mentioned.
Answer: Above sentence has been deleted due to the revision of discussion.
L273 PYCR3 in italic
Answer: Done as requested (Line 266).
Finally, thank you again for the valuable suggestions. If you have any question on this manuscript, please feel free to contact me. Thank you again for your time and favorable consideration.
Sincerely,
Dongxiao Sun, PhD, Professor
Tel/Fax: +86-10-62734653
Email: [email protected]
Reviewer 3 Report
This is a very interesting study. The manuscript adds information on the
genetic mechanisms associated with albumin concentration in dairy cows. I
think the authors should review the manuscript to verify some typos and to
improve some grammatical or writing style errors. I perceived that the
discussion was mainly focused on some specific models, but information on
previous studies in dairy cattle is missing. The discussion would be
improved if previous studies in this topic can be included. I offer the
following specific comments to the authors and hope that they can be
useful in improving the manuscript.
Simple summary (Lines 14 - 23)
I think you can improve the writing of some sentences, e.g., L 14 -15, L 15
-16
Introduction:
L 41 - 42:
Please check this sentence and verify if you can improve the wording.
L 46:
Should urea be considered protein? Please check.
L 50:
Should it be "Previous" instead of "The previous"?
L 52-54:
Please check the sentence wording and style and reword if necessary.
L 54-55:
Please check the sentence wording (use of verbs) and reword.
L 57:
Should it be "is" instead of "are"?
L 62:
Should it be "previous" instead of "precious"?
L 66:
1
Should it be "an" instead of "a"?
L 71:
I suggest you remove the phrase "there is" from the sentence.
Materials and methods
L 77 - 79:
How did you arrive at this sample size? How were the cows selected to be
included in the study?
L 97:
Should it be "Quality control was..." instead of "Quality control were..."?
Results
L 139:
Should it be "are" instead of "were"?
L 143 (Table 1)
Can you please include the unit of measurement of albumin concentration?
Discussion
L 202
Check typo ("resultsin").
Author Response
Dear Reviewer:
Thank you very much for your positive comments and constructive suggestion on the manuscript. We have made the revisions as you suggested.
This is a very interesting study. The manuscript adds information on the genetic mechanisms associated with albumin concentration in dairy cows. I think the authors should review the manuscript to verify some typos and to improve some grammatical or writing style errors. I perceived that the discussion was mainly focused on some specific models, but information on previous studies in dairy cattle is missing. The discussion would be improved if previous studies in this topic can be included. I offer the following specific comments to the authors and hope that they can be useful in improving the manuscript.
Answer: Thanks so much for the Review’s comment and suggestion. We have re-written the discussion as suggested (Lines 209-291).
Simple summary:
(Lines 14 - 23) I think you can improve the writing of some sentences, e.g., L 14 -15, L 15-16
Answer: Thanks so much for the Review’s comment and suggestion. We have re-written the sentences as follows:
The early death and illness of newborn calves result in enormous economic losses in dairy industry. As the immune system has not been fully developed in the neonates, the adequate intake of nutrients and immune substance in colostrum is essential in protecting neonates from infections in their early life (Lines 14-17).
Introduction:
L 41 - 42: Please check this sentence and verify if you can improve the wording.
Answer: Thanks so much for the Review’s comment and suggestion. We have re-written this sentence as follows:
As the immune system of newborn calves is too weak to against various infections, most disease (flu, diarrhea and umbilitis) and death events affecting calves occur in the first few days after birth (Lines 42-44).
L 46: Should urea be considered protein? Please check.
Answer: Thanks so much for the Review’s comment and suggestion. We have deleted urea in the manuscript (Line 46).
L 50: Should it be "Previous" instead of "The previous"?
Answer: Done as requested (Line 51).
L 52-54: Please check the sentence wording and style and reword if necessary.
Answer: We have re-written this sentence as follows:
During the period of mastitis, the mammary gland is exposed to the high level of free radicals, and albumin might enhance anti-oxidant defenses of gland (Lines 53-54).
L 54-55: Please check the sentence wording (use of verbs) and reword.
Answer: We have re-written this sentence as follows:
In dairy cattle, concentration of albumin was found higher in first milked colostrum (1.21 ± 0.44 mg/ml) than it in mature milk (<0.2 mg/ml) (Lines 49-51).
L 57: Should it be "is" instead of "are"?
Answer: Thanks so much for the Review’s comment and suggestion. This sentence has been removed in the revised manuscript.
L 62: Should it be "previous" instead of "precious"?
Answer: Done as requested (Line 57).
L 66: 1 Should it be "an" instead of "a"?
Answer: Done as requested (Line 62).
L 71: I suggest you remove the phrase "there is" from the sentence.
Answer: Done as requested (Line 67).
Materials and methods
L 77 - 79: How did you arrive at this sample size? How were the cows selected to be included in the study?
Answer: Thanks so much for the Review’s comment and suggestion. Firstly, the estimated heritability of 0.27~0.39 for albumin concentration in serum and milk belongs to moderate heritability (Forsblom et al. 1999; Loomis et al. 2019; Luke et al. 2019), so sample size of approximate 600 met the analysis requirements of GWAS. Moreover, to ensure the consistency of physiological conditions and genetic background of cows, we selected cows with following requirements:
(1) Cows were from 10 dairy farms in the Beijing Dairy Cattle Center and the Beijing Sunlon Livestock Development Company Limited, one of the biggest professional breeding company of Chinese Holstein, where the Dairy Herd Improvement system (DHI) has been carried out since 1999 and every cattle has complete pedigree record;
(2) Cows were the offspring of 44 sire families with more than 10 cows per family;
(3) The calving date of cows were relatively closing.
L 97: Should it be "Quality control was..." instead of "Quality control were..."?
Answer: Done as requested (Line 90).
Results
L 139: Should it be "are" instead of "were"?
Answer: Done as requested (Line 131).
L 143 (Table 1) Can you please include the unit of measurement of albumin concentration?
Answer: Yes, of course. We have added the unit of albumin concentration in Table 1.
Discussion
L 202 Check typo ("resultsin").
Answer: Done as requested (Line 220).
Finally, we wish to thank you again for the valuable suggestions. If you have any question on this manuscript, please feel free to contact me. Thank you again for your time and favorable consideration.
Sincerely
Dongxiao Sun, PhD, Professor
Tel/Fax: +86-10-62734653
Email: [email protected]
Round 2
Reviewer 2 Report
The authors have tried to address all of the previous comments and in a short time period, which is well acknowledged. The manuscript has been improved compared to the first version and several parts are presented now in a clearer way. The revised manuscript contains results from another GWAS set-up, where only HD genotyped cows have been analyzed, as sugguested. In this way, imputation error has been avoided, while keeping approximately the same statistical power to detect significant associations. Moreover, the effect of sampling time has been included in the GWAS model to correct for the expected rapid change of albumin concertation after calving.
Despite these improvements, several points still need to be addressed and preclude publication of this study in its present form. Moreover, as previously suggested an extensive English editing is required to improve the quality of this work. The manuscript still contains many grammar and syntax errors. Regarding minor grammar mistakes that have been pointed out in the first revision, unfortunately I have observed that in some cases even though authors replied “Done as suggested” changes have not been made in the manuscript.
Major points that should be addressed are:
- In the original work, 581 and 32 cows genotyped with the 150k and 50k SNP-chip, respectively, were analyzed. In the revised manuscript authors have followed previous suggestions to exclude the imputed cows. However, 572 instead of 581 cows were reported in the revised version. What was the reason for excluding 9 more cows?
- The new GWAS results are entirely different from the original analysis, with only BovineHD2000016546 (BTA20) and BovineHD0700032536 (BTA7) for colostrum and serum albumin concentrations being in common. The significant associations, with few exceptions, appear now in different chromosomes. How this discrepancy can be interpreted? This questions the robustness of the GWAS findings.
- The distribution plots of the measured albumin concentrations reported in Figure 1 are probably wrong and inconsistent with values reported in Table 1. More precisely, the distributions of the measured albumin concentrations (Figure 1a,b) contain negative values; this cannot be possible. Moreover, the mean of log-transformed serum albumin is 1.28. However, based on visual inspection of Figure 1d the values do not match, since 1.28 fits on the right tail of the distribution.
- The SD of the measured colostrum albumin is 0.555 > 0.454 (the mean). A SD higher than the mean questions the quality of the measurement.
Minor comments:
As reported in the first revision, the text contains many syntax errors. I listed again few of these errors below, but this list represents a random subselection only. I suggest once again all authors to proofread the text prior to resubmission.
- L17 replace “Albumin are a multifunctional proteins” with “The term albumin refers to a group of multifunctional proteins”.
- L18 replace “…calves to against…” with “…calves against…”.
- L21 the word “study” is used twice in the same sentence. Please, rephrase.
- L21 replace “Genome-wide” with “genome-wide”.
- L21 if you say genome-wide association study, then it should be was performed and not were performed.
- L29 replace “140,668 Single Nucleotide Polymorphisms (SNPs)” with “containing 140,668 Single Nucleotide Polymorphisms; SNPs”.
- L34 instead of reporting “3 and 12 genes”, you can report the actual gene names (their abbreviations). I think this is important information to the reader in the abstract.
- L34 replace “as the candidates” with “as candidates”.
- L42 replace “objectives” with “objective” – not changed as suggested in the first revision.
- L43 replace “(flu, diarrhea, …)” with “(for e.g., flu, diarrhea, …)”.
- L43 umbilitis. Do you mean omphalitis?
- L51 please rephrase “mature milk”.
- L53 please rephrase “the period of mastitis”. You can simply say “during mastitis”.
- L54 replace “to the high level” with “to thigh level”.
- L57 add space after “trait”.
- L57-58 replace “Previous studies … Heritability estimates…[27-28]” with “Previous studies in human reported heritability estimates of 0.10-0.24 and 0.30-0.39 for glycated and excreted albumin, respectively. In dairy cattle, heritability is relatively lower…[29-30]”.
- L64 replace “detect the genetic” with “detect genetic”.
- L71 here and throughout the manuscript replace “new-born” with “newborn”.
- L80 replace “age” with “old”.
- L85 here and throughout the manuscript replace “log10” with “log10”.
- L96 please rephrase. As far as I know, GCTA does not perform single-marker regression, but fits all SNP simultaneously in the model.
- L104 do not abbreviate GRM since the term appears only once in the manuscript. – not changed as suggested in the first revision.
- L109 replace “caused” with “cause”.
- L110 replace “the threshold p-value” with “the p-value threshold”.
- L114 replace “the totally used SNPs number” with “total number of SNP analyzed”.
- L116 replace “GENABEL” with “GenABEL”.
- L116 you have to cite also the R software (R Core Team. 2013. R: A language and environment for statistical computing. http://www.R-project.org/). – not added as suggested in the first revision.
- L117 you have to cite also the qqman R package (Turner, S. D. 2014. qqman: An R package for visualizing GWAS results using Q-Q and Manhattan plots. bioRxiv https://doi.org/10.1101/005165.) – not added as suggested in the first revision.
- L120 replace “the significant” with “of the significant”.
- L121 the correct link is Ensembl Bos taurus UMD3.1 database http://www.ensembl.org/index.html).
- L131 replace “serum are” with “serum were”.
- L132 replace “phenotype” with “phenotypic”.
- L132 replace “log-transformed” with “log10-transformed”.
- L134 replace “were” with “are”.
- L134 delete “total” – not changed as suggested in the first revision.
- L135 replace “analysis,” with “analysis. The SNPs distribution”
- L135 here and throughout the manuscript be consistent with abbreviation of SNP. Use either SNP or SNPs.
- L138 replace “Distribution” with “distribution”.
- L138 and 139 replace “Indicated” with “show the”.
- L139 replace “log10-transformed serum albumin concentration” with “log10-transformed colostrum and serum albumin concentrations.”
- Table 1
- replace “Standard Deviation” with “SD” and “Coefficient of Variation” with “CV”. Explain them below the Table.
- Use super script a in Traits, i.e. Traitsa. Then use also a when explaining the traits below the Table.
- L142 replace “ser,” with “ser=”.
- Replace “log10, log10-transformed”, with “log10= log10-transformed”.
- L154 replace “~” with “-”.
- L156 and 161 delete “and Figure 2”.
- Table 2
- Use either 2 or 3 decimals after “.”
- Heading of Table 2: please indicate that these are results of the genome-wide association analyses.
- Replace “SEe” with “SEd”.
- Replace “σa2d" with “σa2e”.
- Is the p-value the FDR-corrected p-value? If yes, please stated it like this.
- Please, could you add the proportion of additive genetic variance explained by each significant SNP over the total genetic variance instead of sigma2a?
- Please remove the horizontal line
- Figure 3 please improve the quality. Although the quality is better compared to the previous version still it is not clear.
- L174 replace “-log10-transformed” with “-log10-transformed”.
- L175 replace “Function” with “Functional”.
- In section 3.3. and elsewhere needed in the manuscript, please use “~” as approximation of the gene positions.
- L177 replace “1Mb of up/downstream” with “1Mbp up/downstream”. Please, throughout the manuscript be consistent with abbreviation of Mbp.
- L189 replace “of inflammatory response, immune response” with “of inflammatory and immune response”.
- L192 what do you mean by “interval of known QTLs”?
- Table 3
- do not abbreviate GWAS on the title, use the full name instead.
- remove horizontal line
I will stop here with corrections. As above mentioned, extensive English editing is required.
Round 3
Reviewer 2 Report
I acknowledge the effort of the authors to address the points raised. Regarding the analysis, I do not have anything more to add. However, the manuscript still needs extensive english editing that will improve the quality of this work.
The Discussion can be further amended. Simply reporting the functions of the identified genes do not add to scientific contribution. A concrete story should be made out of the role of the identified genes with a clear conclusion. To achieve this for e.g., subsections could be used to summarize results over a group of genes that share common functionality, such as a common role in immunity. Moreover, reported values of SNP effect, MAF and CGV should be also included in the Discussion. For example, there are SNP with positive and negative effects. In general, information that is included in Tables should be discussed in text.
In Table 3 use comma separator for gene position as used in Table 2.
"Mature milk" (L53) does not exist as a term, please use "milk" instead.
The sum of CGV is ~52%. To me this is most likely an overestimation that should be briefly discussed.
